# Biofortified Rocket (*Eruca sativa*) with Selenium by Using the Nutrient Film Technique

**Carolina Seno Nascimento [1], Camila Seno Nascimento [1], Guilherme Lopes [2], Gilda Carrasco [3], Priscila Lupino Gratão [4] and Arthur Bernardes Cecílio Filho [1,*]**

1    Department of Agricultural Production Sciences, São Paulo State University, Jaboticabal 14884-900, Brazil
2    Department of Soil Science, Federal University of Lavras, Lavras 37200-900, Brazil
3    Department of Horticulture, University of Talca, Talca 3460000, Chile
4    Department of Biology, São Paulo State University, Jaboticabal 14884-900, Brazil
*    Correspondence: arthur.cecilio@unesp.br

**Abstract:** Selenium (Se) is an essential micronutrient for humans, but most foods are Se deficient, mainly because of its low content in the soil. A Se-deficient diet results in increased susceptibility to cardiovascular disease, cancer, and hyperthyroidism. Agronomic biofortification is a good alternative to increase Se in food. This study investigated the effect of Se on the growth, yield, and biofortification of the rocket. Plants were grown in a hydroponic system. Seven Se concentrations (0, 10, 20, 30, 40, 50, and 60 μM) were evaluated using sodium selenate. Growth, yield, lipid peroxidation, hydrogen peroxide content, and the enzymatic activity of catalase and ascorbate peroxidase were influenced by the Se concentration. Considering the evaluated parameters, 10–30 μM Se promoted the best results, and with 20 μM, the higher yield. Rocket plants treated with Se in the nutrient solution were biofortified, showing Se contents of 598.96 to 1437.56 mg kg$^{-1}$ in the dry mass, higher than plants cultivated in a nutrient solution without Se, which presented 167.84 mg kg$^{-1}$ of Se. Se concentrations of 10–30 μM in the nutrient solution were beneficial for rocket plants, while concentrations above 50 μM were toxic to the plants.

**Keywords:** Se deficiency; malnourishment; selenate; soilless cultivation; urban agriculture

## 1. Introduction

Selenium (Se) is an essential micronutrient for humans and animals. It is a constituent of about 25 selenoproteins, which participate in several physiological and biochemical processes [1]. A diet deficient in this micronutrient can lead to numerous health issues, including Keshan disease, cardiovascular disease, cancer, hyperthyroidism, and male infertility [2–5].

Estimates show that around 15% of the world's population has Se deficiency [6]. This situation is probably because Se must be incorporated into human and animal nutrition through agricultural products, which present low levels of the nutrient as a result of cultivation in Se-poor soils [7]. Contents of Se in the soil show great spatial variation (usually from 0.01 to 2 mg kg$^{-1}$), with a world average of 0.40 mg kg$^{-1}$, although there are seleniferous soils with up until 1200 mg kg$^{-1}$ [8,9].

Given this situation of natural deficiency of Se in soils and aiming to increase its concentration in food to benefit the health of the population, fertilization of crops with the target element has been studied, a practice known as agronomic biofortification. This technique is recognized to be effective for increasing the Se content in commercial products, and has also been successful in vegetables [10–16]. Vegetables are promising strategies to help solve hidden hunger [17], since vegetables have good prospects for Se biofortification due to their inherent tendency to accumulate Se [18] and the possibility to cater to population growth, rapid urbanization, and growing health concern [19]. In addition, vegetables are an important source of nutrients for the growing vegetarian and vegan communities.

Therefore, the choice reinforces the need for biofortified vegetables to cater to this part of the population.

Several factors influence the success of Se biofortification, but two major considerations are the crop species and the method of Se fertilization [20]. Species belonging to the Brassicaceae family are typically good Se accumulators and, when biofortified with Se, can produce metabolites functioning as cancer-preventing agents, such as Se-methylselenocysteine (SeMetCys) and -glutamyl-Se-methylselenocysteine [18,21]. Among species of this family, rocket is considered a good vegetable to be biofortified [11,15], because it is able to tolerate high Se concentrations in its tissues up to 1000 mg kg$^{-1}$ dry matter [15]. Rocket is a vegetable rich in sulfur [22], which contributes to the success of biofortification due to the similar chemical and physical properties between Se and sulfur. Hence, selenate is transported across the plasma membrane by high-affinity sulfate transporters [1,10,18]. In addition, its short cycle and easy production are motivating factors for rocket biofortification. Rocket has a pungent or bitter flavor and abundant nutrients (potassium, sulfur, iron, and vitamins A and C) in the edible leaves [23] and glucosinolates and antioxidant compounds, whose pharmaceutical and anticancer properties have been proven [24,25]. Because of these characteristics, it has been a vegetable with increasing demand by the population.

Regarding the fertilization method, some studies have compared soil and foliar supply of Se as agronomic methods for biofortification, and the results showed that foliar supply can be up to eight times more efficient than soil [26]. On the other hand, Sabatino et al. [27] observed that hydroponic cultivation was more effective than the foliar method to biofortify endive with selenium. With rocket species (*Eruca sativa* and *Diplotaxis tenuifolia*), Dall'Aqua et al. [11] evaluated Se concentrations in the nutrient solution grown in pots on the biofortification and S-compounds. The authors observed that leaf fresh and dry weight and root dry weight were not affected by Se concentrations from 5 to 20 μM, which provided maximum Se contents of 100 and 300 mg kg$^{-1}$ for *E. sativa* and *D. tenuifolia*, respectively. On the other hand, when grown with 40 μM Se, phytotoxicity and reduction of biomass were observed in both species. Reductions of S-compounds, such as cysteine, methionine, glucosinolates, and phenolic compounds, were observed in both species.

In the present study, we chose the nutrient film technique (NFT) as a vehicle to biofortify the rocket. This growing system, widely used for the cultivation of leafy vegetables can increase not only yield but also the quality and safety of fresh produce and thus meet the demands of modern society [28]. It also allows greater efficiency in the use of water and nutrients compared with soil cultivation, which has promoted its prominence among leafy vegetable growing systems [29,30]. Biofortification of plants with selenium via hydroponic cultivation may be more efficient than the soil biofortification method due to the absence of interactions of the element with the physical, chemical, and biological attributes of the soil [31–33]; safer; and more effective than the foliar method due to the absence of drift, rain, and equipment calibration [28]; and perhaps more economical due to the lower dose required for biofortification as the roots remain continuously in contact with the nutrient solution.

Parallel to the human health benefits of Se, research has shown that Se can also act beneficially on plant growth and development by positively affecting the cellular antioxidant system, photosynthetic rate, and tolerance to biotic and abiotic stresses [34,35], which may favorably affect crop yield. However, excess Se increases the production of reactive oxygen species, decreases the photosynthetic efficiency and the growth of the plant, causes chlorosis, and can lead to the death of the plant [36,37]. The range between beneficial and toxic levels is narrow [38] and needs to be known because species differ in their ability to accumulate and tolerate Se [39]. Growing *Eruca sativa* in a hydroponic system, Dall'Acqua et al. [11] observed that when the Se concentration in nutrient solution changed from 20 to 40 μM, the Se foliar content caused phytotoxicity. Therefore, the development of phytotechnologies for selenium biofortification requires a thorough understanding of the uptake, translocation, and assimilation processes at the molecular, physiological, and

agronomic levels [40], on which the cultivation method is an important factor modifying the plant response.

Therefore, this work aimed to evaluate the effects of the supply of Se on the growth, nutrition, yield, and biofortification of the rocket in hydroponics according to the nutrient film technique.

## 2. Materials and Methods

### 2.1. Location and Characterization of the Area

The experiment was carried out in a greenhouse using the nutrient film technique hydroponic system at UNESP, Brazil (21°15′22″ S, 48°18′58″ W, at an altitude of 575 m.a.s.l.).

During the experimental period, the average minimum, maximum, and overall temperatures were 20, 39.3, and 34.8 °C, respectively. The average relative humidity was 46%, and the minimum and maximum mean were 45% and 89%, respectively.

### 2.2. Treatments and Experimental Design

Seven Se concentrations (0, 10, 20, 30, 40, 50, and 60 µM) in the nutrient solution were evaluated. The experimental design used was a complete randomized block with three replicates. Sodium selenate was used as the source of Se. The experimental unit consisted of a 1.3 m four-channel workbench and a 100 L tank with a submerged pump (Power Head CX-300 model; flow rate of 1000 L h$^{-1}$). The treatments were applied to the plants in the third cultivation phase, when the seedlings were transplanted to the final cultivation channel.

### 2.3. Installation and Conduction of the Experiment

The rocket cultivar 'Folha Larga' (TopSeed®) was sown in a 2 × 2 × 2 cm block of phenolic foam, previously washed in running water for 10 min (Phase 1). Seven days after sowing, when the seedlings displayed expanded cotyledons, they were transferred to the initial growth PVC channels (5 cm in diameter and 5% incline) in a nutrient film technique hydroponic system (Phase 2). During this period, the supply of nutrient solution to the plants was intermittent, alternating between 15 min with circulation and 15 min without circulation.

The transfer to the final channel (10 cm in diameter) was carried out at the two-leaf stage (Phase 3). The spacing of plants adopted was 0.25 × 0.05 m. In this phase, as well as in phase 2, the nutrient solution proposed by Furlani [41], which is recommended to leafy vegetables, mainly lettuce and rocket in NFT system, was used. However, the concentrations of macronutrients were reduced to 80% of those proposed by Furlani [41], that is, 19.2 (N-NH$_4^+$), 139.2 (N-NO$_3^-$), 31.2 (P), 144.0 (K), 114.0 (Ca), 32.0 (Mg), and 41.6 (S) mg L$^{-1}$. Micronutrients were applied at the concentrations recommended by the author as follows: 0.3 (B), 0.02 (Cu), 2.0 (Fe), 0.4 (Mn), 0.06 (Mo), and 0.06 (Zn) mg L$^{-1}$. The activation of pumps used to recirculate the nutrient solution was controlled by using a timer, starting at 06:00 h and ending at 19:00 h, without interruption.

The pH was maintained between 5.5 and 6.5 using NaOH and H$_2$PO$_4$ to raise and lower the pH, respectively. The nutrient solution was renewed when the electrical conductivity of the treatments reached 75% of the initial electrical conductivity (dS m$^{-1}$). Two renovations were performed during the experimental period. The treatments started 1 week after transferring the seedlings to the final channels.

### 2.4. Characteristics Evaluated

#### 2.4.1. Biometric Analyses

Before harvest, the plant height (H; cm) of 15 randomly selected plants in each plot was measured from the ground surface to the highest end of the last not yet fully extended leaf using a metric ruler. After harvesting these 15 plants, the characteristics evaluated were: number of leaves per plant (LN); leaf area (LA; cm$^2$ per plant): obtained using a LICOR 3100 electronic meter; shoot fresh mass (SFM; g per plant), obtained immediately after harvest by weighing the plant shoots on an electronic scale; shoot dry mass (SDM; g per plant): plant shoots were washed in running water, water plus detergent solution, and

then deionized water. After that, shoots of the 15 plants were dried in a forced circulation oven at 40 °C until constant weight, and the final weight was recorded; for root dry mass (RDM; g per plant), 50 cm of the root system was collected from each channel, totaling 200 cm of root per plot. The roots were washed, dried, and weighed according to the procedure described for the dry mass of shoot.

### 2.4.2. Nutrient and Se Content

Macronutrient (g kg$^{-1}$) and micronutrient (mg kg$^{-1}$) contents in the dry mass of shoots from 15 plants evaluated in item 2.4.1 were determined as described by Miyazawa et al. [42]. The selenium content (μg kg$^{-1}$) was analyzed by graphite furnace atomic absorption spectroscopy (using an Atomic Absorption Spectrometer with Zeeman background correction and EDL lamp for Se; AAnalyst™ 800 AAS, PerkinElmer) in extracts obtained by acid digestion using $HNO_3$, according to the 3051a method of the United States Environmental Protection Agency [43]. For quality assurance and control of Se measurements in the samples, in each batch of acid digestion, a standard reference material from the Institute for Reference Materials and Measurements (white clover—BCR 402) and a blank sample were included and then analyzed for Se. The mean recovery value obtained for the standard material was close to 117% (the mean value detected was 7.84, and the certified value is 6.7 mg kg$^{-1}$), which is considered satisfactory (less than 20% of the variation). Before harvest, in the same 15 plants, which have had the height measured, the sample of tissue was obtained to evaluate the characteristics described in items 2.4.3 to 2.4.9.

### 2.4.3. Carotenoids and Chlorophyll Content

Three leaf discs of the defined area from 15 plants were collected in the central region of the newly developed leaf. The collected material was added to Eppendorf tubes (previously covered with aluminum foil to avoid contact with light) containing 80% acetone. The obtained material was kept under stirring for 48 h at 4 °C. Then, the contents of carotenoids (470 nm), chlorophyll *a* (662 nm), and chlorophyll *b* (645 nm) were determined spectrophotometrically at the wavelengths indicated. Calculations for the determination of the contents of each pigment were performed using the method proposed by Lichtenthaler [44].

### 2.4.4. Lipid Peroxidation

The lipid peroxidation was estimated based on the content of thiobarbituric-acid-reactive substances, as described by Heath and Packer [45]. Malondialdehyde (MDA) content was obtained through spectrophotometric readings at wavelengths between 535 and 600 nm.

### 2.4.5. Hydrogen Peroxide ($H_2O_2$)

$H_2O_2$ content was obtained by the assay proposed by Alexieva et al. [46]. A fresh plant tissue was homogenized with thiobarbituric acid (0.1%); then this material was centrifuged at 10,000 rpm for 10 min. The resultant supernatant was added to a medium containing 100 mM potassium phosphate buffer (pH 7.5) and 1 M potassium iodide. The obtained solution remained in an ice container for 1 h. The reading was performed at 390 nm. $H_2O_2$ concentration was determined by a standard $H_2O_2$ curve.

### 2.4.6. Stress Signaling Enzymes

Before the determination of stress signaling enzymes, the leaves of the rocket, collected at the end of the experiment and stored at −80 °C, were homogenized (2:1 volume of extraction buffer to fresh tissue mass) with potassium phosphate buffer (100 mM, pH 7.5) containing 1 mM ethylenediaminetetraacetic acid (EDTA), 3 mM dithiothreitol, and 5% polyvinylpolypyrrolidone [47]. The homogenate was centrifuged at 10,000 rpm for 30 min, and the supernatant was stored in aliquots at −80 °C for analyzing the antioxidant enzymes.

### 2.4.7. Superoxide Dismutase (SOD, E.C. 1.15.1.1)

SOD activity (U mg$^{-1}$ protein) was determined as proposed by Giannopolitis and Ries [48]. The reaction occurred in a medium consisting of 50 mM potassium phosphate buffer (pH 7.8), 10 mM EDTA, 50 mM methionine, 0.1 mM riboflavin, 1 mM nitrotetrazolium blue chloride, and an aliquot of the enzyme extract.

### 2.4.8. Catalase (CAT, E.C. 1.11.1.6)

The reaction medium consisted of 50 mM potassium phosphate buffer (pH 7.0) and 12.5 mM $H_2O_2$ at a final volume of 3 mL, with the reaction initiated by the addition of 20 μL of extract. The CAT activity (μmol min$^{-1}$ mg$^{-1}$ protein) was determined by monitoring the decomposition of $H_2O_2$ at 240 nm for 1 min [49].

### 2.4.9. Ascorbate Peroxidase (APX, E.C. 1.11.1.11)

APX activity was obtained by a reaction of the plant extract, 5 mM ascorbate, 80 mM potassium phosphate buffer (pH 7.0), 1 mM EDTA, and 1 mM $H_2O_2$ [49]. The reading was obtained by monitoring the oxidation rate of ascorbate at 290 nm at 30 °C for 1 min.

### 2.5. Statistical Analysis

Data were submitted to analysis of variance with an F-test at 5% of probability and a regression study using the AgroEstat program [50]. Significant equations with a higher coefficient of determination were adopted.

## 3. Results

### 3.1. Growth, Yield, and Nutrient and Se Contents

The H, LN, LA, SFM, SDM, RDM, and yield were all influenced by the Se concentration in the nutrient solution. SFM and SDM presented quadratic adjustments in response to Se concentrations (Figure 1A,B).

The maximum SFM (40.19 g per plant$^{-1}$) and SDM (2.8 g per plant$^{-1}$) were obtained with 20 and 24 μM Se, respectively. Amounts of 40 and 49 μM Se, SFM, and SDM, respectively, presented lower values than plants that did not receive Se, noting a toxic effect on plants.

Similar to SFM and SDM were the responses of H, LA, LN, and RDM, which presented maximum values with Se concentrations between 22 and 24 μM (Figure 1C–F). Using a nutrient solution with Se concentrations above 44 μM, Se caused toxicity in the rocket, and the biometric values obtained were lower than those observed for plants that were cultivated in the nutrient solution without Se.

The highest yield (3.78 kg m$^{-2}$) was obtained with 20 μM Se in the nutrient solution and with 941 mg kg$^{-1}$ Se in the SDM (Figure 2). The leaf content of Se presented a quadratic adjustment in response to the Se concentrations in the nutrient solution. The maximum estimated content was 1.443 mg kg$^{-1}$, obtained with 54 μM Se (Figure 2).

None of the micronutrients and, among the macronutrients, only K ($y = 0.4679x + 72.726$; $R^2 = 0.65$; F = 20.12; $p < 0.01$) and S ($y = 0.0181x + 7.4549$; $R^2 = 0.98$; F = 20.99; $p < 0.01$) were influenced by the Se concentrations. Table 1 shows the appropriate leaf nutrient contents [51] and the observed contents. Although they were not influenced by the treatments, the foliar contents of B and Cu were similar to the appropriate ranges, however, with some levels just above and just below the appropriate range, respectively. Fe, Mn, and Zn leaf contents were observed far above the appropriate range. Although these nutrients were not found within the appropriate ranges for rocket, no symptoms of deficiency and toxicity were observed, and according to the analysis of variance, these results are not due to the increase in the concentration of Se in the nutrient solution ($p > 0.05$, test F).

### 3.2. Physiological and Enzymatic Characteristics

Chlorophyll *a*, chlorophyll *b*, and carotenoids were not influenced by the Se concentrations in the nutrient solution.

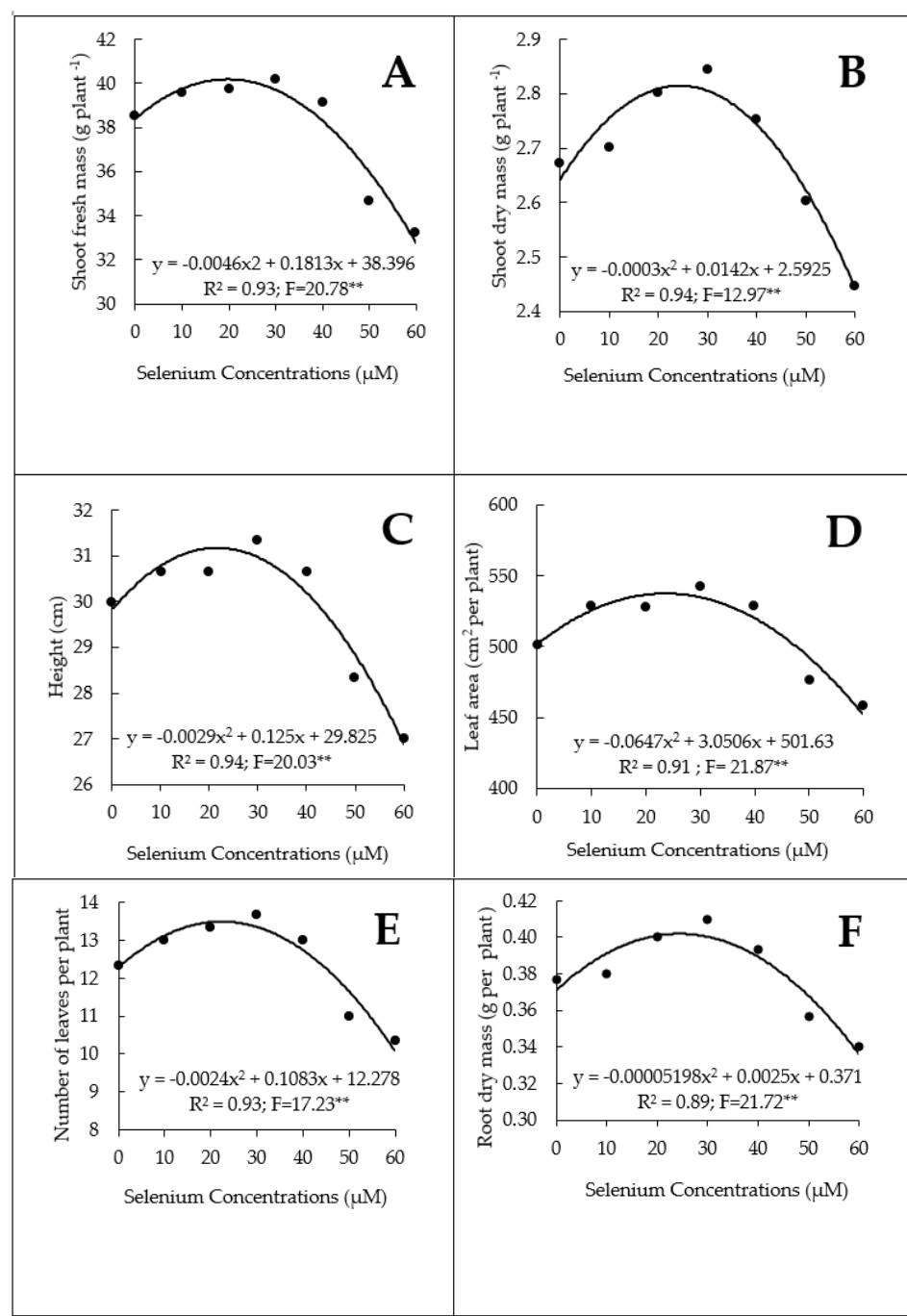

**Figure 1.** Shoot fresh mass (**A**), shoot dry mass (**B**), height (**C**), leaf area (**D**), number of leaves (**E**), and root dry mass (**F**) of rocket plants in response to selenium concentrations. ** Significant at $p \leq 0.01$.

**Table 1.** Adequate (Adeq.) and observed (Observ) values for leaf nutrient content in rocket plant.

| | N | P | K | Ca | Mg | S |
|---|---|---|---|---|---|---|
| | | | g kg$^{-1}$ | | | |
| Observ. | 42.9–46.0 | 4.5–5.0 | 77.0–111.0 | 24.6–26.7 | 6.2–6.8 | 7.4–8.5 |
| Adeq. | 40.0–50.0 | 3.0–8.0 | 30.0–60.0 | 20.0–40.0 | 4.0–7.0 | 4.0–9.0 |
| | B | Cu | Fe | Mn | Zn | |
| | | | mg kg$^{-1}$ | | | |
| Observ. | 74.4–88.7 | 3.7–5.0 | 400.0–466.7 | 166.7–216.7 | 300 | |
| Adeq. | 25.0–60.0 | 5.0–20.0 | 100.0–300.0 | 50.0–160.0 | 45.0–80.0 | |

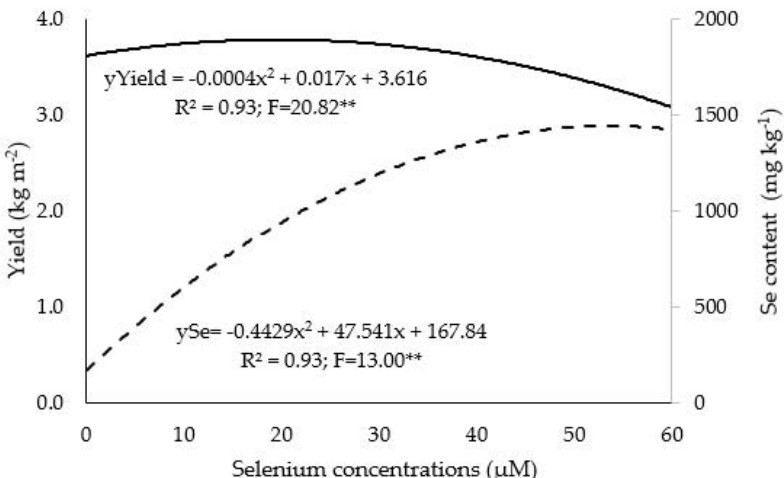

**Figure 2.** Rocket productivity and leaf selenium content in response to selenium concentrations in the nutrient solution. ** Significant at $p \leq 0.01$.

Lipid peroxidation, expressed by the MDA content, showed a significant difference ($p < 0.05$) between the lowest and the highest applied concentrations, with 28.3% less MDA when 10 μM Se was used (Figure 3A). The $H_2O_2$ contents at concentrations of 0 and 10 μM Se were 20.6% and 22.2%, respectively, lower than the content obtained with 60 μM Se (Figure 3B). The SOD activity did not change significantly ($p > 0.05$) with increasing Se concentration (Figure 3C). On the contrary, the CAT activity at concentrations of 20 and 30 μM Se was 41.1% and 40.1% lower, respectively, than the activity with 60 μM Se (Figure 3D). APX activity was higher with 50 μM Se, presenting reductions of 41.9 and 41.5% at Se concentrations of 20 and 30 μM Se, respectively (Figure 3C). Although not presenting a significant difference ($p > 0.05$), the treatments with 10, 20, 30, and 40 μM Se presented lower APX activity than plants that did not receive Se.

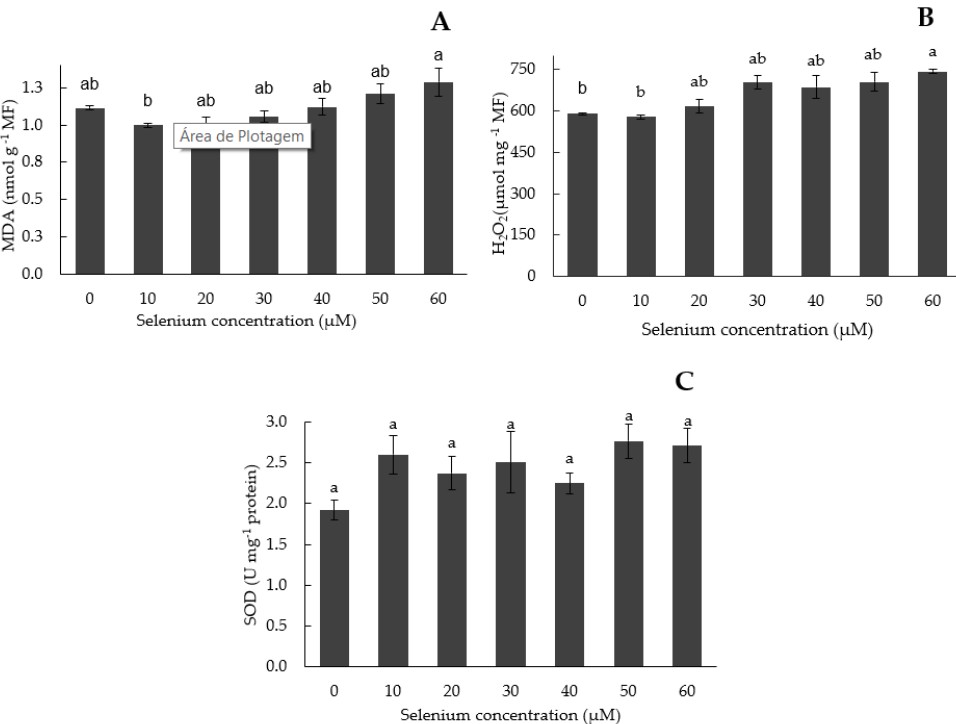

**Figure 3.** *Cont*.

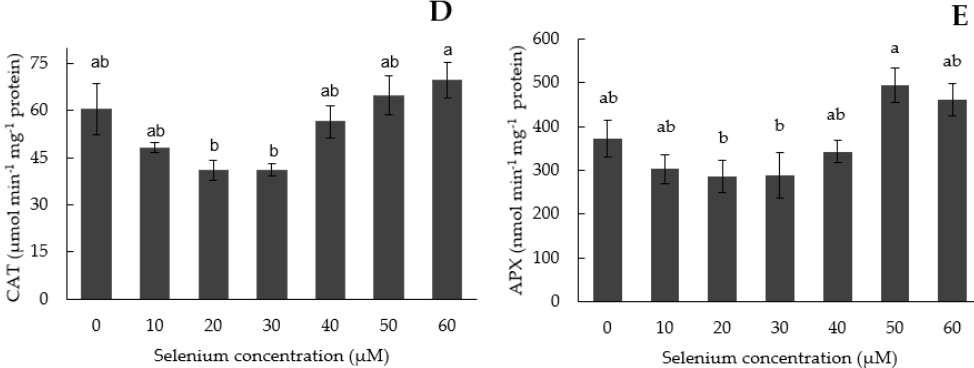

**Figure 3.** Effect of selenium concentration on lipid peroxidation (MDA) in rocket leaves (**A**). Effect of selenium concentration on the hydrogen peroxide content ($H_2O_2$) in rocket leaves (**B**). Effect of selenium concentration on superoxide dismutase (SOD) in rocket leaves (**C**). Effect of selenium concentration on catalase (CAT) in rocket leaves (**D**). Effect of selenium concentration on ascorbate peroxidase (APX) in rocket leaves (**E**). The values represent the means of each treatment (*n* = 3). Means followed by the same lowercase letter do not differ by Tukey's test (*p* > 0.05). Vertical bars designate standard error (±EP).

## 4. Discussion

### 4.1. Growth

All evaluated agronomic characteristics showed increases when the plant was cultivated with low Se concentrations, more precisely between 10 and 30 μM Se. The beneficial effect of low Se concentration on the biometric characteristics of rocket plants led to an increase in yield that was maximized with 20 μM Se in the nutrient solution and 941 mg kg$^{-1}$ Se in the leaf dry mass. The results differ from those obtained by Dall'Acqua et al. [11], who did not observe an increase in leaf dry mass of *Eruca sativa* and *Diplotaxis tenuifolia* when up to 20 μM of Se was applied in a hydroponic system, deep water technique. With the same 20 μM Se in the nutrient solution, these authors observed 100 and 300 mg kg$^{-1}$ Se dry mass in the two species, respectively. On the other hand, when Zafeiriou et al. [16] grew rocket in calcareous soils, the authors observed that the best dry mass was obtained with 5 mg kg$^{-1}$ as selenite and 18.5 mg kg$^{-1}$ dry mass.

At 10 to 30 μM Se in the nutrient solution, the lowest MDA, $H_2O_2$, APX, and CAT values were observed, which may be explained by the role of Se in stimulating the antioxidant activity of the cells, positively reflecting on photosynthetic activity and, consequently, on the rocket growth. The by-product of lipid peroxidation, namely, MDA, is used to measure the damage caused by ROS to cell membranes [52]. High concentrations of $H_2O_2$ (ROS) negatively affect plant growth and development. APX and CAT act to decompose $H_2O_2$ to $H_2O$ and $O_2$, thereby reducing the production of the highly toxic hydroxyl radical [53]. In addition to increased antioxidant activity, other direct effects of Se on carbohydrate content, stomatal conductance, photosynthetic pigments, and electron transport rate are also cited [54,55].

At concentrations above 30 μM Se, the leaf Se contents, MDA, and $H_2O_2$ were increased, as well as the activities of APX and CAT enzymes. Therefore, the higher the leaf Se content, the greater the levels of stress indicators. Se toxicity in plants has been attributed to its action promoting oxidative stress. The overproduction of ROS causes injuries to different parts of the plant, such as the cell membrane, DNA, RNA, and proteins [56], causing mutations and loss of structure and function, which irreversibly damages plant growth and development, leading to a decline in crop production.

In the present study, APX and CAT showed high activity at the same concentrations of Se in which the highest amounts of $H_2O_2$ were noted, characterizing enhanced action of the cellular antioxidant system to prevent the formation of ROS. However, even with the highest activity of the enzymes, ineffectiveness and cell damage were verified by

lower growth and yield of the rocket under high Se concentrations in the nutrient solution (>50 μM Se), characterized by both high levels of MDA and $H_2O_2$.

With 50 μM Se in the nutrient solution, the leaf content of Se reached 1.437 mg kg$^{-1}$, which caused rocket toxicity, with a 5% reduction in plant dry mass. The observed result corroborates those of other authors, although differences were observed regarding the critical values of Se in the nutrient solution and leaf tissue. Dall'Acqua et al. [11] observed phytotoxicity of Se in *Eruca sativa* and *Diplotaxis tenuifolia* when Se contents in leaf dry mass reached 200 and 300 mg kg$^{-1}$, which caused reduction in leaf fresh and dry mass and root dry mass for both species. Zafeiriou et al. [16] observed a reduction in leaf dry mass when the Se content was higher than 50 mg kg$^{-1}$. With only 5 and 10 mg kg$^{-1}$ selenate in calcareous soils, the Se contents reached 695.9 and 1070.5 mg kg$^{-1}$, which caused a 6.5 and 84.5% reduction in leaf dry mass, respectively. Ríos et al. [57] and Ríos et al. [58] found an approximate reduction of 5% and 4%, respectively, in lettuce dry matter when applying 80 μM Se to the nutrient solution and foliar contents of approximately 36 mg kg$^{-1}$ Se in both studies. In cucumber, Hawrylak-Nowak et al. [59] observed that low Se concentrations (6–20 μM) promoted the growth of the shoot and root system, but when submitted to 80 μM, the plants presented 648 and 603.5 mg kg$^{-1}$ Se, respectively, and had a reduction of 15% in the shoot and 21% in the root.

In general, the increase in Se supply did not cause major changes in the arugula's nutritional status, except for S and K. The high foliar levels of micronutrients in rocket, which were not motivated by the concentrations of Se in the nutrient solution, may be due to the high temperatures in the experimental period, which led us to preventively reduce the concentration of macronutrients by 20%, justifying that such reduction also should have been applied to micronutrients. Due to the chemical similarity between Se and S, Se relies on sulfate carriers for absorption [60]. The linear growth of S observed with the increase in Se concentrations can be explained by the synergistic relationship between these two nutrients. Similar results are observed in the literature studies in which the supply of Se increased the S content [61]. Rocket is a species of the Brassicaceae family, which is known to have a high demand for S [62]. High levels of S promote plant development because this nutrient is responsible for the formation of disulfide bonds in the essential amino acids cysteine and methionine, which confers structural stability to proteins, favoring growth and production [63]. In addition to its structural function, S also has important enzymatic functions in photosynthesis and respiration [64].

The effect of Se on K increase may have contributed to the enhanced control of water loss by this vegetable, favoring metabolism even under high-temperature conditions. K presents itself as an important osmotic compound, so proper supply is paramount for proper stomatal movement and cell elongation. Besides, K facilitates the diffusion of $CO_2$ from the atmosphere to chloroplast [65] and is involved in phloem mediation, photoassimilate distribution, and sucrose transport [66].

Thus, the direct and indirect actions of Se collaborated to promote, in low concentrations in the nutrient solution, rocket growth and productivity relative to plants that did not receive Se in the nutrient solution.

### 4.2. Se biofortification in Hydroponic Systems

The nutrient solution containing 20 μM Se, which resulted in the highest yield, provided an increase of 461% (168–941 mg kg$^{-1}$) of Se in rocket leaves compared with plants that did not receive Se. This fact shows that this species has a great affinity to accumulate Se.

According to the Australian and New Zealand Code of Dietary Standards [67], the maximum allowable limit for Se in foods is 1 mg kg$^{-1}$ dry mass. Thus, for the rocket to be marketed, it must be cultivated in nutrient solution with a maximum of 22 μM Se. This concentration allows for the maximum permitted in vegetable biofortification, and with an estimated yield loss of only 0.06% to the maximum obtained. At 23 μM Se and above, the Se content in food is higher than that established by the standards and is not recommended for biofortification programs aimed at in natura consumption of the product.

The German, Austrian, and Swiss nutrition societies have set benchmarks for daily intake of Se as a function of age [68]. Considering the 999 mg kg$^{-1}$ Se content in the dry mass of rocket grown with 22 μM Se, and that the rocket plant contains 93% water, then the fresh mass content of the rocket is equivalent to 70 mg kg$^{-1}$ Se. Assuming the consumption of 50–100 g of rocket in an adult meal, this biofortified vegetable will provide 4–7 mg Se, which is 5%–10% of a man's daily requirement and 5.9%–11.8% of a woman's daily need.

## 5. Conclusions

Se concentrations of 10–30 μM Se in the nutrient solution favor the growth of the rocket, characterized by lowered stress indicator levels and lowered antioxidant enzyme activities.

Se concentrations above 50 μM Se and a minimum content of 1437 mg kg$^{-1}$ Se cause toxicity in the rocket plants and reduce its dry mass by 5%.

At maximum yield with 20 μM Se supplementation of the nutrient solution, biofortification of rocket plants is observed, attaining 461% enrichment in the Se content compared with plants not cultivated in nutrient solution supplemented with Se (168–941 mg kg$^{-1}$).

**Author Contributions:** Conceptualization, A.B.C.F. and G.C.; methodology, A.B.C.F.; formal analysis, C.S.N. (Carolina Seno Nascimento) and C.S.N. (Camila Seno Nascimento); investigation, C.S.N. (Carolina Seno Nascimento), C.S.N. (Camila Seno Nascimento), G.L., and P.L.G.; resources, A.B.C.F. and G.L.; data curation, C.S.N. (Carolina Seno Nascimento), C.S.N. (Camila Seno Nascimento), G.L., P.L.G., and A.B.C.F.; writing—original draft preparation, C.S.N. (Carolina Seno Nascimento), C.S.N. (Camila Seno Nascimento), P.L.G., and G.C.; writing—review and editing, A.B.C.F.; supervision, A.B.C.F.; project administration, A.B.C.F. All authors have read and agreed to the published version of the manuscript.

**Funding:** This research received no external funding.

**Data Availability Statement:** Not applicable.

**Acknowledgments:** To CNPq and CAPES.

**Conflicts of Interest:** The authors declare no conflict of interest.

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
