# Peer review of "Biofortified Rocket (Eruca sativa) with Selenium by Using the Nutrient Film Technique"

_horticulturae, doi:10.3390/horticulturae8111088_

Round 1

Reviewer 1 Report (Previous Reviewer 4)

The authors considerably improved the manuscript compared to the first submission. However, a number of aspects still require improvements before the paper can be published. 

Specific suggestions and recommendations for improvement:

L89-91: the absence of drift, rain, equipament calibration [28] and perhaps more economical due to the lower dose requeired for biofortification as the roots remain continuously in contact with the nutrient solution

Please check spelling: equipament – equipment; requeired – required. It is recommended to run a spell check throughout the entire text.

L106-107: Therefore, this work aimed to evaluate the effects of the supply of Se on the growth, nutrition, yield, and biofortification of the rocket.

To be more specific and in line with aspects discussed in the introduction section, the method of application (in a hydroponic system) should be mentioned in the aim of the study.

L133-137: In this phase, as well as phase 2, the nutrient solution used was proposed by Furlani[42], which is recommended for leafy vegetables, mainly lettuce and rocket in NFT system. The concentrations used were: 19.2 (N-NH4 + ), 139.2 (N-NO3 - ), 31.2 (P), 144.0 (K), 114.0 (Ca), 32.0 (Mg) and 41.6 (S) mg L-1 , corresponding to 80% of the original macronutrient concentrations.

Grammar! The first sentence should read: “In this phase, as well as phase 2, the nutrient solution proposed by Furlani[42], which is recommended to leafy vegetables, mainly lettuce and rocket in NFT system, was used.

However, the second sentence indicates that the authors only used reduced concentrations of macronutrients corresponding to 80% of the concentrations proposed by Furlani. If this is the case, the sentence should be clarified to avoid confusion for the readers. Possible modification: “In this phase, as well as in phase 2, the nutrient solution proposed by Furlani [42], which is recommended for leafy vegetables, mainly lettuce and rocket in NFT system, was used. However, the concentrations of macronutrients were reduced to 80% of those proposed by Furlani [42], i.e. 19.2 (N-NH4+), 139.2 (N-NO3-), 31.2 (P), 144.0 (K), 114.0 (Ca), 32.0 (Mg) and 41.6 (S) mg L-1.

L149-150: Before harvest, the plant height (H; cm) was were measured 15 plants randomly selected in each plot, from the ground surface to the highest end of the leaf, not extended, using a metric ruler.

Grammar! Suggested edit: “Before harvest, plant height (H; cm) of 15 randomly selected plants in each plot was measured from the ground surface to the highest end of the last, not yet fully extended leaf, using a metric ruler.”

L272-275: None of the micronutrients, and among the macronutrients, only K (y = 0.4679x + 72.726; R² = 0.65; F = 20.12; p < 0.01) and S (y = 0.0181x + 7.4549; R² = 0.98; F = 20.99; p < 0.01) were influenced by the Se concentrations. Table 1 shows the appropriate leaf nutrient contents [52] and the observed contents.

Looking at Table 1, the statement for K among the macronutrients is correct. However, the observed contents of S are within the appropriate leaf nutrient contents (S: 7.4 – 8.5 observed values versus 4.0 -9.0 adequate values).

Based on Table 1, some of the micronutrients have also been affected by the Se treatments, although the authors stated that they were not affected by the Se treatments. The observed values for B. Fe, Mn, and Zn are considerably higher than the stated adequate ones, while the content of Cu is slightly lower compared to the indicated adequate range. Please make corresponding adjustments to the respective statement in the manuscript.

L345: Zafeiriou et al. [16] observed a redution in leaf dry mass

Spelling error: “reduction”

L353: the plants presented 648 and 603.500 mg kg-1 Se

One decimal digit appears to be sufficient in this case: 603.5 mg kg-1 Se

L355-356: In general, the increase in Se supply did not cause major changes in the arugula's nutritional status, except for S and K.

This statement should be corrected as micronutrient content was also affected based on Table 1 (please see earlier comment regarding this aspect).

L375: 4.2. Agronomic biofortification

Like the adjustments made in the title of the manuscript, this section heading should also be adjusted to, for example, “Se biofortification in hydroponic systems”.

L376-378: The nutrient solution containing 20 µM Se, which resulted in the highest yield, provided an increase of 461% (168–941 mg kg-1 ) of Se in the rocket leaf than in plants that did not receive Se.

Grammar! Suggested edit: “The nutrient solution containing 20 µM Se, which resulted in the highest yield, provided an increase of 461% (168–941 mg kg-1 ) of Se in rocket leaves compared to plants that did not receive Se.”

L400-402: At maximum yield, there is agronomic biofortification of the rocket, attaining 461% enrichment in the element’s content than plants not cultivated in nutrient solution with Se (168–941 mg kg-1 ).

Grammar! Suggested edit: “At maximum yield with xx µg Se supplementation of the nutrient solution, biofortification of rocket plants is observed, attaining 461% enrichment in the Se content compared to plants not cultivated in nutrient solution supplemented with Se (168–941 mg kg-1 ).”

Author Response

Dear Reviewer #1

We thank you for again dedicating some of your time to improving our manuscript. We apologize for such gross mistakes in the English language, which could not have occurred.

The responses to your observations and requests follow. In the text, we highlight the corrections in red.

L89-91: The request has been fulfilled.

L106-107: The request has been fulfilled.

L133-137: Thank you very much for contributing to better writing about what we had said. We accepted your suggestion and the wording was redone.

L149-150: The request has been fulfilled.

L272-275: This time we understand what the Reviewer is telling us. Failure to comply with the request in the previous opportunity was not a matter of simply not complying. Undoubtedly, the levels of some nutrients are outside the range of levels suitable for rocket according to the authors. What we said previously and now is that when doing the analysis of variance of the data, the F test is not significant and, therefore, the treatments (Se concentrations) did not influence the contents of these nutrients. It is important for the Reviewer to understand that when we increase the Se concentration in the nutrient solution, there is no effect on the nutrient content, that is, the variation observed in the treatment means is not significant at 5% probability by the F test, i.e. the variation is not due to Se concentrations. Thus, we express the average content of these nutrients. Our statement, therefore, is correct and there is no way to change it. However, we agree with the Reviewer that it is important to add that the contents of some nutrients are above or below the range suitable for rocket, but not caused by the increased availability of Se in the nutrient solution. Finally, we have included in the same paragraph a text that explains what we are trying to say and that contemplates what the Reviewer pointed out, that is, that there are nutrients that are outside the appropriate range, which is not due to the increase in concentrations of Se.

L345: The request has been fulfilled.

L353: The request has been fulfilled.

L355-356: The request has been fulfilled.

L375: The request has been fulfilled.

L376-378: The request has been fulfilled.

L400-402: The suggestion was accepted and the wording of the last conclusion was revised. In order to do so, we eliminated the penultimate conclusion that we had put forward, which dealt with the concentration of Se in the nutrient solution that maximized productivity. In fact, it got much better.

THANK YOU VERY MUCH FOR YOUR PATIENCE AND COLLABORATION

Reviewer 2 Report (Previous Reviewer 3)

Dear Authors: Now that I have seen the responses to the comments I made in the review, I am pleased with the answers and changes made by them, except for one aspect.  I think in the comments of Table 1, as they explained to me, authors should add a phrase in which they should state: "Contents of Fe, Mn, Zn are above
the appropriate level for rocket due to the micronutrient concentrations in the nutrient solution and the climate". Thus everything will be clear to all readers.  The rest is correct and I agree with the publication of this article.

Author Response

Dear Reviewer #2

We thank you for again dedicating some of your time to improving our manuscript.

The response to your observation follow. In the text, we highlight the corrections in red from all three Reviewers.

This time we understand what the Reviewer is telling us. Failure to comply with the request in the previous opportunity was not a matter of simply not complying. Undoubtedly, the levels of some nutrients are outside the range of levels suitable for rocket according to the authors. What we said previously and now is that when doing the analysis of variance of the data, the F test is not significant and, therefore, the treatments (Se concentrations) did not influence the contents of these nutrients. It is important for the Reviewer to understand that when we increase the Se concentration in the nutrient solution, there is no effect on the nutrient content, that is, the variation observed in the treatment means is not significant at 5% probability by the F test, i.e. the variation is not due to Se concentrations. Thus, we express the average content of these nutrients. Our statement, therefore, is correct and there is no way to change it. However, we agree with the Reviewer that it is important to add that the contents of some nutrients are above or below the range suitable for rocket, but not caused by the increased availability of Se in the nutrient solution. Finally, we have included in the same paragraph a text that explains what we are trying to say and that contemplates what the Reviewer pointed out, that is, that there are nutrients that are outside the appropriate range, which is not due to the increase in concentrations of Se. Also, in Discussion item, we added text to explain about high leaf contents of micronutrients.

THANK YOU VERY MUCH FOR YOUR PATIENCE AND COLLABORATION

Reviewer 3 Report (Previous Reviewer 2)

This revised and resubmitted MS has addressed all my previous comments.  

Author Response

Dear Reviewer#3

We thank you for again dedicating some of your time to improving our manuscript. 

This manuscript is a resubmission of an earlier submission. The following is a list of the peer review reports and author responses from that submission.

Round 1

Reviewer 1 Report

General comments

In this descriptive study, the authors investigated the effect of Se on the growth and agronomic biofortification of Se. I think the experimental design is reasonable, and the paper is generally well-written and structured. Although this study has no much novelty. I have not found any new scientific advancement in this study. Moreover, the study is not hypothesis-driven. The objectives can be easily re-written as a scientific hypothesis regarding doses of Se. I suggest that the authors make this change and reformulate all the other sections oriented for these hypotheses.

 Line to line comments

Section 2.  This species can be easily grown from seeds and cultivated, hence the authors must clearly justify why hydroponics has been chosen in this investigation for growing rocket.

Section 2.3. Please clarify when the plants were subjected to Se treatments.  Also please clarify how you decided the concentration of the treatments.

Section 2.1. The Source of seeds must be clarified.

Section 2.3. lines 82-83. How did the author choose the concentrations of micronutrients in the nutrient solution?

Section 2.3. line89. Why author didn’t use HCl as a regular acid to control the pH of the nutrient solution?

Section 2.4. How much tissues were extracted? These must be clearly written at all times and in all places of M&M.

Section 2.4.2 lines 109-113. The QA/QC (quality assurance/quality control) procedures for acid digestion are missing.  For example, what standard reference material was used to quantify the efficiency of acid digestion? What was the efficiency of the acid digestion? Were reagent blanks analyzed?  What were the detection limits of the AAS? Precision and accuracy tests are not reported.

Section 3.1. lines 174, 177,…. Please convert µg Kg-1 to mg Kg-1. Please do the same for all numbers above three decimal places in the whole manuscript.

Section 4. Please avoid using discussion as result section. Please use simply sentences regarding your findings, and then make the possible explanation for discussing your finding. Avoid re-writing the findings again herein. And, please if possible, do not cite the tables or figures in the discussion section (if you do not find this comment appropriate, ignore it).

Author Response

Dear Reviewer, Thank you for your comments and attention devoted to our manuscript.

Response to REVIEWER #1

Section 2.  This species can be easily grown from seeds and cultivated, hence the authors must clearly justify why hydroponics has been chosen in this investigation for growing rocket.

Authors: We chose this arugula production system because it is a growing system of growing use worldwide and, in particular, in Brazil, since it allows for increased productivity, better quality of the vegetable and greater efficiency in the use of inputs. Arugula is second only to lettuce in this cultivation system. Small text on justification of use was added as requested.

Section 2.3. Please clarify when the plants were subjected to Se treatments.  Also please clarify how you decided the concentration of the treatments.

Authors: The Se treatments were applied to the plants when these were transplanted to final channel of culivation. The following text was added to the manuscript, item 2.2: “The treatments were applied to the plants in the third cultivation phase, when the seedlings were transplanted to the final cultivation channel.” For better understanding, in item 2.3, phases 1, 2 and 3 of cultivation were specified. We have moved the information about the composition of the nutrient solution used to the next paragraph.

Our study was conceived in 2013-2014 and, therefore, the Se concentrations evaluated were chosen based on study of Ramos et al. (2010): Ramos S.J., Faquin V., Guilherme L.R.G., Castro E.M., Ávila F.W., Carvalho G.S., Bastos C.E.A., Oliveira C. (2010): Selenium biofortification and antioxidant activity in lettuce plants fed with selenate and selenite. Plant Soil Environ., 56: 584-588.

Section 2.1. The Source of seeds must be clarified.

Authors: The information was added to the manuscript, in item 2.3.

Section 2.3. lines 82-83. How did the author choose the concentrations of micronutrients in the nutrient solution?

Authors: This is the nutrient solution recommended to roket in NFT system, in Brazil. The information was added to the manuscript in the same sentence where we describe nutrient concentrations.

Section 2.3. line89. Why author didn’t use HCl as a regular acid to control the pH of the nutrient solution?

Authors: We prefer to use phosphoric acid instead of hydrochloric acid, because in addition to reducing the pH of the nutrient solution, it is a source of macronutrient (P), as we know that we should avoid adding Cl to the nutrient solution

Section 2.4. How much tissues were extracted? These must be clearly written at all times and in all places of M&M.

Authors: We have provided the information in item 2.4 about the request presented.

Section 2.4.2 lines 109-113. The QA/QC (quality assurance/quality control) procedures for acid digestion are missing.  For example, what standard reference material was used to quantify the efficiency of acid digestion? What was the efficiency of the acid digestion? Were reagent blanks analyzed?  What were the detection limits of the AAS? Precision and accuracy tests are not reported.

Authors: You are right. We apologize that we have forgotten to add this information before.We have used SRM (white clover) and this information was missing in the previous version. Now, we included all the information and the mean recovery was close to 117% (less than 20% of variation). This is now described in the section 2.4.2, as you can see in the revised manuscript, which is: “Macronutrient (g kg-1) and micronutrient (mg kg-1) contents in the dry mass of shoots from 15 plants evaluated in item 2.4.1 were determined as described by Miyazawa et al. [17]. The selenium content (µg kg 1) was analyzed by graphite furnace atomic absorption spectroscopy (using an Atomic Absorption Spectrometer with Zeeman background correction and EDL lamp for Se; AAnalyst™ 800 AAS, Perkin Elmer) in extracts obtained by acid digestion using HNO3, according to the 3051a method of the United States Environmental Protection Agency [18]. For quality assurance and control of Se measurements in the samples, in each batch of acid digestion, a standard reference material from the Institute for Reference Materials and Measurements (White Clover - BCR 402) and a blank sample were included and then analyzed for Se. The mean recovery value obtained for the standard material was close to 117 % (mean value detected was 7.84 and the certified value is 6.7 mg kg-1), which is considered satisfactory (less than 20% of variation).”

Section 3.1. lines 174, 177,…. Please convert µg Kg-1 to mg Kg-1. Please do the same for all numbers above three decimal places in the whole manuscript.

Authors: Ok. We carried out the exchange of the unit, and we replace the figure 2 by one with the Y2 axis in mg kg-1.

Section 4. Please avoid using discussion as result section. Please use simply sentences regarding your findings, and then make the possible explanation for discussing your finding. Avoid re-writing the findings again herein. And, please if possible, do not cite the tables or figures in the discussion section (if you do not find this comment appropriate, ignore it).

Authors: We appreciate your observation. We try to readjust the manuscript as per your request.

Thank you for your comments and attention devoted to our manuscript.

Reviewer 2 Report

This MS investigated the effects of exogenous Se supplication on the growth, nutrition, yield, and Se contents of Rocket. The results and conclusion of this MS are solid and it should be suitable for publishing in Horticulturae. Minor comments to this MS: The positions of Figure 1E and F in the main text should be adjusted to be consistent with 1A to 1D.  The plants treated with none Se also showed a content of 167.84 ug kg-1 Se in the dry mass, what are the potential reasons for the phenomenon?

Author Response

Dear Reviewer, Thank you for your comments and attention devoted to our manuscript.

Response to REVIEWER #2

This MS investigated the effects of exogenous Se supplication on the growth, nutrition, yield, and Se contents of Rocket. The results and conclusion of this MS are solid and it should be suitable for publishing in Horticulturae. Minor comments to this MS: The positions of Figure 1E and F in the main text should be adjusted to be consistent with 1A to 1D.  The plants treated with none Se also showed a content of 167.84 ug kg-1 Se in the dry mass, what are the potential reasons for the phenomenon?

Authors: We appreciate your comments. The Se concentration in the control is very, very low, and we believe that it is a possible contaminant present in fertilizers.

Thank you for your comments and attention devoted to our manuscript.

Reviewer 3 Report

The idea behind this research is good, to increase the Se concentrations in vegetables (rocket), because much Se necessary for a healthy life comes from animal meat/matter and people who adopt a vegetarian diet do not have at their disposal so many foods rich in Se to cover the daily necessities. So producing vegetables with a richer content in Se comes to their aid.

Regarding the research paper, Title and Abstract clearly state the objective of this experiment.

Throughout the paper, authors are speaking of 7 concentrations of Se which they tested (0, 10, 20, 30, 40, 50, 60) but then they speak of 22, 24, 44 µM. I ask the authors to please explain this.

Sufficient details regarding methods/process are provided so that another researcher is able to reproduce the experiments described.

Authors did provide relevant and current references during discussion.

In the Results section, in the comments for Table 1, the authors state that only K and S have been influenced by Se concentrations and none of the microelements. But from Table 1, it can be seen that Fe, Mn and Zn have higher values than expected.

In the Conclusion section of the paper, I dont understand the first phrase, it must be a mistake.

Conclusions are correct, based on actual results and figures and summarize the results obtained in the experiment.

Author Response

Dear Reviewer, Thank you for your comments and attention devoted to our manuscript.

Response to REVIEWER #3

The idea behind this research is good, to increase the Se concentrations in vegetables (rocket), because much Se necessary for a healthy life comes from animal meat/matter and people who adopt a vegetarian diet do not have at their disposal so many foods rich in Se to cover the daily necessities. So producing vegetables with a richer content in Se comes to their aid.

Authors: Thank you for your comment.

Regarding the research paper, Title and Abstract clearly state the objective of this experiment.

Authors: Thank you for your comment.

Throughout the paper, authors are speaking of 7 concentrations of Se which they tested (0, 10, 20, 30, 40, 50, 60) but then they speak of 22, 24, 44 µM. I ask the authors to please explain this.

Authors: The values cited were obtained from the polynomials equations presented in the  figures.

Sufficient details regarding methods/process are provided so that another researcher is able to reproduce the experiments described.

Authors: Thank you for your comment.

Authors did provide relevant and current references during discussion.

Authors: Thank you for comment.

In the Results section, in the comments for Table 1, the authors state that only K and S have been influenced by Se concentrations and none of the microelements. But from Table 1, it can be seen that Fe, Mn and Zn have higher values than expected.

Authors: The Reviewer is correct in noting that the Fe, Mn and Zn contents are above the appropriate level for rocket. It may be due to the micronutrient concentrations in the nutrient solution and the climate. But, as we said, the concentration of Se (treatments) only affected K and S, for which we show the polynomial equations.

In the Conclusion section of the paper, I dont understand the first phrase, it must be a mistake.

Authors: The Reviewer is correct. Our apologies for the error. We deleted the phrase.

Conclusions are correct, based on actual results and figures and summarize the results obtained in the experiment.

Authors: Thank you for your comments and attention devoted to our manuscript.

Reviewer 4 Report

Summary

This manuscript deals with Se biofortification of rocket grown in hydroponic systems. Se concentrations of 10-30 µM in the nutrient solution were beneficial for the growth of rocket plants and their Se content, while concentrations above 50 µM showed toxic effects.

Specific suggestions and recommendations

L2 – Title: As rocket plants were not grown in the field but in hydroponic systems, a slightly modified title is recommended, such as “Hydroponic biofortification of rocket with selenium” or “Biofortification of rocket with selenium under soilless cultivation”

L21: The best results were obtained with Se concentrations between 20 and 29 µM.

As only concentrations of 0, 10, 20, 30, 40, 50, and 60 µM Se in the nutrient solution were tested, the indicated range between 20 and 29 µM is not valid. It should either be a range of 20 to 30 µM or 20 µM only.

What do you mean by the term “best results”? Please be specific. Do you refer to the Se concentration in the final produce or other parameters?

L53-55: The increasing demand for the rocket, is a short cycle and its easy production are motivating factors for rocket biofortification.

Grammar! Suggested edit: “The increasing demand for rocket, its short cycle, and easy production are motivating factors for rocket biofortification”.

L55-56: In hydroponic systems, its growth is accelerated by the high availability of water and nutrients, with a glimpse of the successful absorption and use of Se by plants.

Style! Suggested edit: “In hydroponic systems, its growth is accelerated by the high availability of water and nutrients, facilitating the absorption and use of Se by plants”.

L62-63: (21°15'22” S, 48°18'58” W, and an altitude of 575 62 m.a.s.l.).

Suggested edit: “(21°15'22” S, 48°18'58” W, at an altitude of 575 62 m.a.s.l.).

L79-80: In this phase, the nutrient solution used was proposed by Furlani [16], with concentrations…

The sentence is not entirely clear. Suggested edit: “ In this phase, the nutrient solution proposed by Furlani [16] was slightly modified and used with concentrations…”

L97-98: 15 plants randomly selected in each plot before harvest were measured from the ground surface to the highest end of the foil, not extended, using a metric ruler;

The term “foil” is not clear in this context. Do you mean “leaflet”?

L103: After that, it was dried in a forced circulation oven at 40 ºC until constant weight

The word “it” is not clear. Suggested edit: “After that, the shoots were dried in a forced circulation oven at 40 ºC until constant weight”

L211-214: None of the micronutrients, and among the macronutrients, only K (y = 0.4679x + 211 72.726; R² = 0.65; F = 20.12; p < 0.01) and S (y = 0.0181x + 7.4549; R² = 0.98; F = 20.99; p < 212 0.01) were influenced by the Se concentrations. Table 1 shows the appropriate leaf nutrient contents [26] and the observed contents.

The statement for K among the macronutrients is correct. However, based on Table 1, some of the micronutrients have also been affected by the Se treatments. The observed values for B. Fe, Mn, and Zn are considerably higher than the stated adequate ones, while the content of Cu is slightly lower compared to the indicated adequate range. Please make corresponding adjustments to the respective statement in the manuscript.

L274-275: Ríos et al. [32] and Ríos et al. [33] found an approximate reduction of 5 and 4%,

Please correct as follows: “Ríos et al. [32,33] found an approximate reduction of 5 and 4%,

L320-321: This section is not mandatory but can be added to the manuscript if the discussion is unusually long or complex.

This sentence should be deleted as the authors decided to add a Conclusion section.

L328-330: At maximum yield, there is agronomic biofortification of the rocket, attaining 461% enrichment in the element’s content than plants not cultivated in nutrient solution with Se (941.5–167.8 µg kg-1 ).

The sentence is unclear; this statement needs to be improved for easy understanding.

Suggested edit: “At maximum yield with xx µg Se supplementation of the nutrient solution, Se enrichment of rocket plants reaches 461% of the value observed in plants grown in a hydroponic system without Se supplementation (167.8-941,5 µg kg-1 Se).

Author Response

Dear Reviewer, Thank you for your comments and attention devoted to our manuscript.

Response to REVIEWER #4

L2 – Title: As rocket plants were not grown in the field but in hydroponic systems, a slightly modified title is recommended, such as “Hydroponic biofortification of rocket with selenium” or “Biofortification of rocket with selenium under soilless cultivation”

Authors: We accepted the suggestion for the title and it has been modified.

L21: The best results were obtained with Se concentrations between 20 and 29 µM.

As only concentrations of 0, 10, 20, 30, 40, 50, and 60 µM Se in the nutrient solution were tested, the indicated range between 20 and 29 µM is not valid. It should either be a range of 20 to 30 µM or 20 µM only.

What do you mean by the term “best results”? Please be specific. Do you refer to the Se concentration in the final produce or other parameters?

Authors: We appreciate the Reviewer's comment and agree with it. We changed the range to 10-30 µM when referring to all parameters and specified 20 µM as the best Se concentration for the yield. In this way, we believe that the wording has become clearer. The new range is in accordance with the results presented and the range reported in the Discussion and Conclusion items.

L53-55: The increasing demand for the rocket, is a short cycle and its easy production are motivating factors for rocket biofortification.

Grammar! Suggested edit: “The increasing demand for rocket, its short cycle, and easy production are motivating factors for rocket biofortification”.

Authors: Thank you for your suggestion. We accepted and modified the wording.

L55-56: In hydroponic systems, its growth is accelerated by the high availability of water and nutrients, with a glimpse of the successful absorption and use of Se by plants. Style! Suggested edit: “In hydroponic systems, its growth is accelerated by the high availability of water and nutrients, facilitating the absorption and use of Se by plants”.

Authors: Thank you for your suggestion. We accepted and modified the wording.

L62-63: (21°15'22” S, 48°18'58” W, and an altitude of 575 62 m.a.s.l.).

Suggested edit: “(21°15'22” S, 48°18'58” W, at an altitude of 575 62 m.a.s.l.).

Authors: Thank you for your suggestion. We accepted and modified the wording.

L79-80: In this phase, the nutrient solution used was proposed by Furlani [16], with concentrations…

The sentence is not entirely clear. Suggested edit: “ In this phase, the nutrient solution proposed by Furlani [16] was slightly modified and used with concentrations…”

Authors: We made corrections to this information, including to serve another Reviewer. We believe it is clear.

L97-98: 15 plants randomly selected in each plot before harvest were measured from the ground surface to the highest end of the foil, not extended, using a metric ruler;

The term “foil” is not clear in this context. Do you mean “leaflet”?

Authors: Thank you for attention. No, no, “leaflet” is not appropriate. We exchange by “leaf”.

L103: After that, it was dried in a forced circulation oven at 40 ºC until constant weight

The word “it” is not clear. Suggested edit: “After that, the shoots were dried in a forced circulation oven at 40 ºC until constant weight”

Authors: Thank you for your suggestion. We accepted and modified the wording.

L211-214: None of the micronutrients, and among the macronutrients, only K (y = 0.4679x + 211 72.726; R² = 0.65; F = 20.12; p < 0.01) and S (y = 0.0181x + 7.4549; R² = 0.98; F = 20.99; p < 212 0.01) were influenced by the Se concentrations. Table 1 shows the appropriate leaf nutrient contents [26] and the observed contents.

The statement for K among the macronutrients is correct. However, based on Table 1, some of the micronutrients have also been affected by the Se treatments. The observed values for B. Fe, Mn, and Zn are considerably higher than the stated adequate ones, while the content of Cu is slightly lower compared to the indicated adequate range. Please make corresponding adjustments to the respective statement in the manuscript.

Authors: The Reviewer is correct in noting that some content of nutrients are above the appropriate level for rocket. It may be due to the micronutrient concentrations in the nutrient solution and the climate. But, as we said, the concentration of Se (treatments) only affected K and S, for which we show the polynomial equations

L274-275: Ríos et al. [32] and Ríos et al. [33] found an approximate reduction of 5 and 4%,

Please correct as follows: “Ríos et al. [32,33] found an approximate reduction of 5 and 4%,

Authors: Thank you for your suggestion. We accepted and modified the wording.

L320-321: This section is not mandatory but can be added to the manuscript if the discussion is unusually long or complex.

This sentence should be deleted as the authors decided to add a Conclusion section.

Authors: Thank you for your observation. We deleted the phrase. Our apologies for the error.

L328-330: At maximum yield, there is agronomic biofortification of the rocket, attaining 461% enrichment in the element’s content than plants not cultivated in nutrient solution with Se (941.5–167.8 µg kg-1).

The sentence is unclear; this statement needs to be improved for easy understanding.

Suggested edit: “At maximum yield with xx µg Se supplementation of the nutrient solution, Se enrichment of rocket plants reaches 461% of the value observed in plants grown in a hydroponic system without Se supplementation (167.8-941,5 µg kg-1 Se).

Authors: Thank you for your suggestion. But we prefer to keep the wording as it is because this sentence is a conclusion, and we do not want let it too long. However, the Reviewer can see that the same information is presented in Discussion item, where we show the complete information about the Se conncentration to obtain the highest yield.

Round 2

Reviewer 1 Report

Although the manuscript appears to have been revised with care, in my judgment, the authors couldn’t address the raised issues. Besides, the manuscript is not novel and the data is not strong enough to assure publication in the journal “Horticulturae”.

Author Response

Dear Associate Editor and Reviewer - Horticulturae, MDPI

We thank you for your attention to our manuscript. We have accepted your recommendations and implemented improvements in the title and introduction item.

In the title, we make it more strictly connected to what we have studied, including the requested words. As now the scientific name appears in the title (as requested), we have eliminated it from the key words and included two other words (Se-deficience and Urban agriculture).

In the Introduction item, the improvements are explained in red in the new version of the manuscript and correspond to: a) better understanding about the issue of vegetable biofortification (beginning of the third paragraph and fifth paragraph); b) addition of text to better explain why we use hydroponic cultivation to biofortify arugula (third paragraph) and c) addition of text about the importance of arugula (end of the fifth paragraph). To this end, 11references were added, which are from journals with high impact factor (Agronomy, Agriculture, Frontiers in Plant Science, Arch. Agron. Soil Sci, Scientia Hortic., Innovare J. Agric. Sci., and Horticulturae. In time, we inform you that we have made the necessary corrections in the item References and the connections with their respective citations in the text.

We are very grateful for the requests made. We acknowledge that the changes in the wording have improved the understanding of the text and the quality of the manuscript.

The rest of the text has been kept as presented in the latest version of the manuscript.

Thus, we hope to have complied with the requests and hope that our manuscript will be accepted for publication in this respected journal. However, we remain at your disposal to provide any other corrections that may be necessary.

Best regards.
